# Photo-Durable Molecularly Oriented Liquid Crystalline Copolymer Film based on Photoalignment of *N*-benzylideneaniline

**DOI:** 10.3390/polym15061408

**Published:** 2023-03-12

**Authors:** Gento Nakajima, Mizuho Kondo, Moritsugu Sakamoto, Tomoyuki Sasaki, Hiroshi Ono, Nobuhiro Kawatsuki

**Affiliations:** 1Department of Applied Chemistry, Graduate School of Engineering, University of Hyogo, 2167 Shosha, Himeji 671-2280, Japanmizuho-k@eng.u-hyogo.ac.jp (M.K.); 2Japan Science and Technology Agency (JST), Core Research for Evolutional Science and Technology (CREST), Chiyoda-ku, Tokyo 102-0076, Japan; sakamoto@vos.nagaokaut.ac.jp (M.S.); sasaki_tomoy@vos.nagaokaut.ac.jp (T.S.); onoh@vos.nagaokaut.ac.jp (H.O.); 3Department of Electrical Engineering, Nagaoka University of Technology, 1603-1 Kamitomioka, Nagaoka 940-2280, Japan

**Keywords:** liquid crystalline polymer, photoalignment, photoinduced reorientation, birefringent film, *N*-benzylideneaniline

## Abstract

Copolymer films of photoalignable liquid crystalline (LC) copolymethacrylates comprised of a phenyl benzoate mesogen connected with *N*-benzylideneaniline end moiety (NBA2) and benzoic acid (BA) side groups exhibit a photoinduced reorientation behavior. Significant thermally stimulated molecular reorientation attains a dichroism (*D*) greater than 0.7 for all copolymer films and a birefringence of 0.113–0.181. In situ thermal hydrolysis of the oriented NBA2 groups decreases the birefringence to 0.111–0.128. However, the oriented structures of the film are maintained, demonstrating a photo-durability, even though the NBA2 side groups photo-react. The hydrolyzed oriented films show higher photo-durability without changing their optical properties.

## 1. Introduction

Molecularly oriented films using photoalignable liquid crystalline polymers (PLCPs) can be directly applied to birefringent optical films and diffraction devices [1,2,3,4,5]. Photoalignment films combined with thermally-curable materials on the photoalignment layer are used to fabricate linear and/or patterned birefringent elements [6,7], and many types of polarization grating are investigated based on the photoalignment of photosensitive polymeric materials [8,9,10,11,12,13,14]. Axis-selective photoisomerization, photo-crosslinking, and/or photo-rearrangement reactions of photosensitive materials upon exposure to linearly polarized (LP) light form molecularly oriented structures [15,16,17,18,19,20,21]. Among them, cinnamate derivative-containing polymer shows a small optical anisotropy after axis-selective photo-crosslinking [22,23,24,25], but the anisotropy is amplified at an elevated temperature due to thermally induced self-organization when the material indicates LC characteristics [15,19]. Additionally, azobenzene and *N*-benzylideneaniline (NBA) containing polymeric films show large photoinduced molecular reorientations based on axis-selective *trans-cis-trans* photoisomerization [16,17,18,20]. Thermally stimulated amplification of the photoinduced optical anisotropy is also observed in these polymeric materials when they exhibit LC characteristics. For practical applications in display and diffraction optical elements, the photoaligned film must show a photo-durability to hold their optical properties.

We have systematically studied the photoinduced reorientation of PLCPs with NBA derivatives side groups (Figure 1a) [20,26]. Although the axis-selective *trans-cis* photoisomerization combined with thermal stimulation produces significant molecularly oriented films [26,27,28], re-exposing the film to UV light alters the oriented structure of the film because the photosensitive moieties often remain in the molecularly oriented film [27]. This feature, especially in azobenzene-containing PLCPs, has potential in devices with a reversible orientation [29,30,31,32,33,34,35,36]. In this context, several studies improve the stability of the oriented structure based on detaching the H-bonded photosensitive moieties after the photoalignment process and introducing the crosslinkable moieties in the oriented films [37,38], while decomposition of the photosensitive moieties in the reoriented film can realize photo-durability.

Recently, we investigated photoinduced reorientation and thermal hydrolysis of PLC copolymers with NBA1 (Figure 1a) and benzoic acid (BA) side groups [26]. When the exposed film is annealed at LC temperature range of the material, significant cooperative molecular orientation both side groups is achieved based on the photoinduced reorientation of NBA1 moieties. Additionally, thermally generated hydrolysis of NBA1 occurs to form non-photoreactive phenyl aldehyde end moieties under high humidity as the BA side groups act as an acid catalyst. The hydrolyzed film shows photo-durability due to the lack of photosensitive parts. However, the composition of NBA1 side groups must be limited to maintain the oriented structure because the hydrolyzed NBA1 side groups do not exhibit LC characteristics. Namely, the oriented structure collapses after the hydrolysis when the content of NBA1 is large. To avoid this disadvantage, photoalignable NBA-containing mesogen (NBA2 in Figure 1a) is designed, which exhibits LC characteristics after the hydrolysis, and a PLCP with NBA2 side groups has been synthesized to explore the LC characteristics of oriented films after the hydrolysis [28]. Although the hydrolysis requires immersing the reoriented film in an acetic acid solution due to lack of acidic moieties, the LC characteristics remain in the hydrolyzed NBA2 side groups. Therefore, a copolymer comprised of NBA2 and BA side groups should exhibit photoinduced molecular reorientation, and subsequent thermal hydrolysis of NBA2 may be realized without a solution-immersing process.

Herein, we synthesize PLC copolymers with NBA2 and BA side groups (**PNx**, Figure 1b), and we investigated their thermally stimulated photoinduced cooperative reorientation. Similar to the homopolymer with NBA2 side groups and the copolymer with NBA1 and BA side groups [26,28], significant cooperative molecular reorientation is achieved. However, the annealing conditions for the thermally stimulated cooperative molecular reorientation after the photo-exposure depend on the content of the BA side groups. For all copolymer compositions, annealing under high humidity conditions generates in situ hydrolysis of the reoriented films while maintaining the molecular oriented structure at a proper temperature. Furthermore, the molecularly oriented structure is preserved upon re-exposing to 365-nm light, which induces a photoreaction of the NBA side groups. Consequently, a higher photo-durability is realized without changing the optical properties of the hydrolyzed film.

## 2. Experimental

### 2.1. Materials

All starting materials were used as received from TCI. Methacrylate monomers comprised of NBA2 and BA side groups were synthesized according to the literature [19,27]. Synthetic procedure of a methacrylate monomer with a 4-formylphenylbenzoate side group is as follows: into a solution of 1.99 g (6.5 mmol) of a methacrylate with BA side groups and 1.19 g (9.7 mmol) of 4-hydroxybenzaldehyde in 25 mL of dichloromethane, a solution of 1.9 g (9.2 mmol) of *N*,*N*’-dicyclohexylcarbodiimide (DCC) and a small amount of *N*,*N*-dimethyl-4-aminopyridine (DMAP) in 25 mL of dichloromethane was added and stirred at room temperature for 23 h. Then, the solution was filtered, and the solvent was evaporated under reduced pressure. The monomer was purified by a column chromatography (silica gel, chloroform eluent) and recrystallized from chloroform. Yield: 2.6 g (5.5 mmol), 85 mol%. M.p. = 58–61 °C.

Copolymers were synthesized by free radical copolymerization of corresponding methacrylate monomers in THF solution (10 wt/vol-%) using α,α-azobis(isobutyronitrile) (AIBN) as the initiator (3 mol-% to the monomers) at 55 °C for one day. After the polymerization, the (co)polymer was purified by reprecipitating several times from a THF solution to methanol and diethyl ether. Then, the monomer feed ratio controlled copolymer composition was adjusted. Table 1 summarizes the composition, molecular weight, and thermal properties of the copolymers.

### 2.2. Photoreaction

Spin-coating onto quartz or CaF_2_ substrates from a THF solution prepared thin copolymer films (100–180-nm thick). The films were photo-irradiated with a light intensity of 30 mW/cm^2^ at 365-nm by a high-pressure Hg lamp equipped with a glass plate placed at Brewster’s angle and a bandpass filter (Asahi Spectra, Tokyo, Japan; REX-250) at room temperature. The films were subsequently annealed at elevated temperatures under a dry N_2_ atmosphere to thermally stimulate molecular reorientation.

### 2.3. Hydrolysis of the Oriented Films

Annealing at elevated temperatures under high humidity (RH > 70%) induced in situ hydrolysis of the oriented film. To attain high humidity at high environmental temperature, an oriented film was placed on a temperature-controlled stage placed in a box with high-temperature water. The annealing time and its temperature-controlled degree of hydrolysis (DH) were adjusted, and DH was evaluated by the change in the absorption band of NBA2 at 335 nm.

### 2.4. Characterization

^1^H-NMR spectra acquired by a Bruker DRX-500 FT-NMR and FT-IR spectra (JASCO, Tokyo, Japan; FTIR-6600) confirmed the copolymers. The molecular weight was measured by a gel permeation chromatography GPC; JASCO, Tokyo, Japan; PU-2080 with a Shodex column using THF as an eluent) calibrated using polystyrene standards. The thermal properties were examined using a polarized optical microscope (POM; Olympus, Tokyo, Japan; BX51) equipped with a Linkam TH600PM heating and cooling stage, as well as a differential scanning calorimetry (DSC; Hitachi High Technologies, Tokyo, Japan; DSC-7200). As a measure of the photoinduced optical anisotropy, polarization absorption spectra were measure using a Hitachi U-3010 spectrometer equipped with Glan-Taylor polarization prisms. The absorption spectrum in both parallel and perpendicular direction to the polarization (**E**) of the LP 365-nm was measured. The photoinduced in-plane dichroism (*D*) is estimated as
*D* = (*A*_⏊_ − *A*_||_)/(*A*_⏊_ + *A*_||_)(1)
where *A*_||_ and *A*_⏊_ are the absorbances parallel and perpendicular to **E** of LP light, respectively.

Birefringence of the 120–150 nm- thick oriented films was measured by a multi-channel polarization analyzer using Optipro-compact11MQ (Shintech Co., Ltd., Yamaguchi, Japan) at 607 nm.

## 3. Results and Discussion

### 3.1. Thermal and Spectroscopic Properties of the Copolymers

The synthesized copolymers exhibit nematic LC characteristics, which are confirmed by the POM observation (Figure 2a). Because the H-bonded dimer of BA is responsible for the LC characteristics of a polymethacrylate with BA side groups (**PBA**) [19,39,40,41,42], PBA has a large transition enthalpy (29.3 J/g) and a LC–isotropic transition temperature (T_i_) of 179 °C. H-bond formation induces LC characteristics in several types of composite materials [43,44,45]. By contrast, the homopolymer with NBA2 side groups (**PN100**) displays a small transition enthalpy and high T_i_, at which simultaneous decomposition of the material is seen in POM observation (Figure 2b, Table 1). For copolymers **PN10** and **PN20**, T_i_ appears at a temperature similar to **PBA** with a large transition enthalpy, suggesting that the LC characteristics disappear due to H-bond cleavage of the BA side groups. Free BA side groups disorder the nematic orientation of NBA2 side groups. By contrast, **PN50** shows the first transition with small enthalpy at 163 °C and a second one at 205 °C. In this case, the LC characteristics are maintained even after cleaving the H-bonded dimer in the BA side groups (N_2_ temperature range). A large amount of NBA2 side groups can maintain the nematic LC characteristics even though the existence of free BA moieties.

UV-vis spectra of the copolymer depend on the copolymer composition. They show the absorption band of BA at 262 nm and that of NBA2 at 335 nm (Figure 2c). Because the absorption band at 262 nm overlaps with the NBA2 absorption, absorption maxima slightly shift to the longer wavelength region as the NBA2 composition increases, while the absorption maxima at 335 nm rarely change due to no absorption of BA at 335 nm. FT-IR spectra show the vibration intensities of C=N (1624 cm^–1^), COOH (1683 cm^–1^), and H-bonded dimer (2681 and 2530 cm^–1^) also depend on the copolymer composition (Figure 2d). These results indicate the random distribution of mesogenic side groups in the copolymers.

### 3.2. Axis-Selective Photoreaction of the Copolymer Films

Exposing the copolymer films to LP 365-nm light generates axis-selective *trans-cis-trans* photoisomerization of the NBA2 side groups [27,28]. Figure 3a–c show the changes in the polarized absorption spectra of **PN10**, **PN20**, and **PN50** films. The inset plots the absorbances at 262 and 335 nm as a function of the exposure energy. For all copolymers, *A*_||_ (*A*_⏊_) decreases (increases) as the exposure energy increases, and a negative photoinduced optical anisotropy (*A*_||_ − *A*_⏊_ < 0) appears due to the photoinduced reorientation of NBA2. The photoinduced optical anisotropy depends on the composition of NBA2 in the copolymer. Negative optical anisotropy is clearly seen in **PN20** and **PN50** films due to large content of NBA2, but it is very small for **PN10**.

Additionally, absorbances for both directions at 335 nm decrease when the exposure energy exceeds 50 J/cm^2^, indicating a side photoreaction of the NBA2 [20]. Although the photoinduced optical anisotropy increases as the exposure energy increases, side photoreaction, such as photodegradation and photo-crosslinking of NBA moieties, inhibits the reorientation [46,47,48].

### 3.3. Thermal Amplification of the Photoinduced Optical Anisotropy

In case of the photoinduced molecular reorientation of LC polymeric films, thermal stimulation of the exposed films often amplifies the photoinduced optical anisotropy, where the amplification direction depends on the type of the material [1,15]. When the photoinduced reorientation is caused by the axis-selective photoisomerization, thermal amplification often occurs in the same direction of the photoinduced optical anisotropy, but in some cases, out-of-plane direction is sometimes amplified [26,49,50,51]. Significant in-plane thermal stimulation of the photoinduced optical anisotropy was observed in the PLCP with NBA1 and BA side groups [29]. Similarly, annealing the exposed copolymer film in its LC temperature range produces a significant cooperative molecular orientation in a direction perpendicular to the polarization of LP 365-nm light. Figure 4a–c show the change in the polarized absorption spectra of **PN10**, **PN20**, and **PN50** films before and after exposure to LP 365-nm light for 5 J/cm^2^ and subsequent annealing at elevated temperatures. Photoinduced small anisotropy (*D*_355_ < 0.1) is significantly amplified after the annealing, where the cooperative molecular reorientation is generated for all copolymer films. *D*_262_ (*D*_335_) values of the reoriented films are 0.71 (0.71) for **PN10**, 0.80 (0.79) for **PN20**, and 0.80 (0.86) for **PN50**. The birefringence values of these films are 0.113 (**PN10**), 0.158 (**PN20**), and 0.181 (**PN50**) at 607 nm. The birefringent value increases as the amount of NBA2 groups increases due to the higher inherent birefringence of NBA2, where the birefringence of the reoriented **PN100** film is 0.24 [27,28].

A similar thermal stimulation is observed when the exposure energy is 100 J/cm^2^ (Figure 4d–f). In these cases, photoinduced optical anisotropy of NBA2 is larger than those for 5 J/cm^2^—dose films—while *D*_262_ (*D*_335_) values of the thermally amplified reoriented films are 0.69 (0.63) for **PN10**, 0.66 (0.75) for **PN20**, and 0.77 (0.81) for **PN50**, which are similar to those obtained from the 5 J/cm^2^-dose films. However, copolymers reveal different suitable range of the exposure energy for a sufficient cooperative molecular reorientation (Figure 4a–c, inset). For **P50**, thermal stimulation (*D* > 0.7) is generated when the exposure energy is between 1 and 200 J/cm^2^. **P20** films exhibit smaller exposure range between 2 and 100 J/cm^2^, but **P10** has a much smaller range of 5 and 30 J/cm^2^ for *D* > 0.6. When the composition of the NBA2 is low, the power for the cooperative thermal stimulation is small in the total mesogenic side groups.

### 3.4. Influence of the Annelaing Temperature

The annealing temperature influences the cooperative molecular reorientation. Figure 5a,b plots the change in the thermally stimulated *D* values of the exposed copolymer films as a function of the annealing temperature when the exposure energy is 5 J/cm^2^ (100 J/cm^2^). When the exposure energy is 5 J/cm^2^, the highest temperature for the significant *D* values of **PN10** (**PN20**) is 165 °C (160 °C), which is close to T_i_ of the copolymers. By contrast, **P50** exhibits a siginifcant thermal amplification in the N_1_-LC temperature range, but it does not exhibit coopertative reorientation when the annealing temperature is in the N_2_-LC temperature range. Cooperative reorientation occurs when the annealing temperature is 120 °C (N_1_ temperature range) because BA side groups reveal H-bonded dimer showing LC characteristics (Figure 6a). However, axis-selectively oriented NBA2 side groups cannot align free-BA side groups at 170 °C (Figure 6b), where the BA side groups do not form H-bonding at the N_2_ temperature range.

Furthermore, Figure 5 indicates that the highest temperatures for the reorientation in all films with a 100 J/cm^2^ dose are lower than those with a 5 J/cm^2^ dose. For **PN10**, a significant thermal stimulation occurs when annealing at 165 °C for an exposed film for 5–30 J/cm^2^ (Figure 4a and Figure 7a), but the orientated structure is not amplified at 165 °C for the film with a 100 J/cm^2^ dose (Figure 7b). Similarly, a significant thermal stimulation appears in **PN20** when annealing at 160 °C for an exposed film for 5 J/cm^2^, but reorientatiom does not occur for the film with a 100 J/cm^2^ dose (Figure 7c,d). These results indicate that the side-photoreaction of the NBA2 side groups at the high doses decreases T_i_ of the film after the photoreaction, which is confirmed by the POM observation.

### 3.5. Hydrolysis of the Reoriented Film

The oriented C=N bond in the copolymer film is hydrolyzed under humid conditions at elevated temperatures. Schiff base derivatives are hydrolyzed under the acid condition [52,53]. In case of the copolymers, BA side groups serve as the acid for the hydrolysis [26], whereas hydrolysis of reoriented **PN100** requires a treatment with an acetic acid solution due to lack of the acidic moieties [28]. For the thermal hydrolysis of NBA2 in the copolymer, 4-methoxy-phenylamine formed upon the hydrolysis sublimes simultaneously upon the annealing.

Figure 8a–c respectively show the changes in the polarized absorption spectra of reoriented **PN10**, **PN20**, and **PN50** films before and after hydrolysis when the films are annealed at 140 °C (**PN10**) or 130 °C (**PN20**, **PN50**). The insets plot the changes in the absorbances (normalized *A*_⏊_ at 262 and 335 nm) as a function of the hydrolysis time for various annealing temperatures. All films display a diminished absorbance at 335 nm when the annealing time exceeds 60 min (degree of the hydrolysis >95%). However, the absorbance at 262 nm rarely changes, indicating that the reoriented structure is maintained after the hydrolysis (*D*_262_ = 0.80, 0.67, and 0.78 for **PN10**, **PN20**, and **PN50**, respectively). This is because the LC characteristics of the oriented film are maintained, indicating that the isotropic transition temperature (T_i_) hydrolyzed film is higher than the annealing temperature. To confirm this prediction, copolymers of methacrylate monomers with a 4-formylphenylbenzoate and BA side groups (**PAx**) are synthesized (Figure 9a). These copolymers show nematic LC characteristics, and their T_i_ is plotted in Figure 9b. Although T_i_ of **PAx** is lower than that of **PNx** due to the lack of NBA moiety, thermal hydrolysis temperature is low enough to hold the LC characteristics. Additionally, the hydrolyzed-oriented films display decreased birefringent values of 0.111, 0.141, and 0.128 for **PN10**, **PN20**, and **PN50** at 607 nm, respectively, as the C=N decomposition in the NBA2 side groups to form phenyl aldehyde end moieties decreases the inherent birefringence.

The hydrolysis rate and stability of the orientated structure depend on the hydrolysis temperature. Although the hydrolysis occurs faster as operating at the higher annealing temperature (Figure 8a–c, insets), oriented structure for **P10** maintains even when the annealing at 160 °C while gradually collapses for **P20** at 150 °C. However, the oriented structure of **P50** collapses within 30 min when annealed at 150 °C (Figure 10a–c). This is because T_i_ of the oriented film gradually decreases upon the hydrolysis, while T_i_ of the complete-hydrolyzed film becomes lower than the annealing temperature (Figure 9b). Assuming hydrolysis is a first order reaction, and the Arrhenius plot of the hydrolysis of the copolymer films (Figure 8d) indicates that the activation energies of the **PN10**, **PN20**, and **PN50** are 87.6, 78.1, and 64.0 kJ/mol, respectively. These values are comparable as compared to the hydrolysis of phenyl ester derivatives [54].

### 3.6. Photo-Durability of Reoriented Films

Re-exposing an oriented **PN100** film to LP 365-nm light in a direction parallel to the orientation direction reverses the molecular orientation direction [27]. This is due to the axis-selective photoreaction of NBA2 side groups, which is similar to the behavior of azobenzene-containing polymeric films [29,30,31,32,33,34,35,36]. By contrast, a different photo-durability behavior is observed for the oriented copolymer films.

Figure 11a–c, respectively, show the changes in the polarized absorption spectra of oriented **PN10**, **PN20**, and **PN50** films before and upon the exposure to LP 365-nm light with **E** parallel to the orientation direction. The insets plot the *D*_262_ and *D*_335_ values as a function of the exposure energy. For **PN10** and **PN20**, the absorption bands at 262 nm in both directions rarely change even when the re-exposure energy exceeds 100 J/cm^2^, whereas the band at 335 nm gradually decreases. Thus, the molecularly orientated structure does not collapse even though the photoreaction of the NBA2 side groups proceeds in the parallel direction. Namely, photoreaction of NBA2 does not generate the cooperative molecular reorientation of mesogenic side groups. In this case, the birefringence of the film decreases slightly to 0.112 (**PN10**) and 0.157 (**PN20**). By contrast, the molecularly orientated structure of **PN50** collapses slightly (Figure 11c). The absorption band at 262 nm shows a slight decrease in the reoriented structure. After the 100 J/cm^2^ dose, *D*_262_ and the birefringence values decrease to 0.61 and 0.132, respectively. These results indicate that a larger composition of the oriented BA side groups restricts the photoinduced molecular reorientation upon the re-exposure at room temperature. Namely, the cooperative motion of the mesogenic side groups is negligible once the reoriented structure is formed, whereas photodegradation of NBA2 partly occurs at high exposure doses.

### 3.7. Photo-Durability of Hydrolyzed Oriented Films

Hydrolyzed oriented films exhibit a higher photo-durability. Figure 12a–c shows the changes in the polarized absorption spectra of a hydrolyzed oriented **PN10**, **PN20**, and **PN50** films (DH > 95%) upon the exposure to LP 365-nm light with **E** parallel to the orientation direction, respectively. The spectra of the oriented films do not change even after the exposure dose of 100 J/cm^2^, regardless of the composition. This is due to the lack of photo-sensitive moieties in the oriented films. These films are applicable to birefringent film for display application because there is no absorption in the visible light region.

## 4. Conclusions

Photoalignable liquid crystalline copolymers comprised of NBA2 and BA side groups are synthesized. Exposing these copolymers films to LP 365-nm light generates photoinduced reorientation of NBA2 side groups, and subsequent annealing in the LC temperature range of the material induces significant cooperative molecular orientation of both mesogenic side groups. However, thermal amplification of the cooperative orientation does not occur when the H-bond of the BA dimer collapses, even though the copolymer exhibits LC characteristics. The oriented films exhibit birefringent values of 0.113–0.181 at 607 nm, and the oriented structure is maintained under LP 365-nm light re-exposure when the NBA2 content is less than 20%. Because in situ hydrolysis of the NBA2 moiety does not alter the oriented structure, the resultant films display a higher photo-durability without affecting the optical properties due to the lack of photosensitive moieties. Currently, studies to apply these copolymers to birefringent devices are underway.

## Figures and Tables

**Figure 1 polymers-15-01408-f001:**
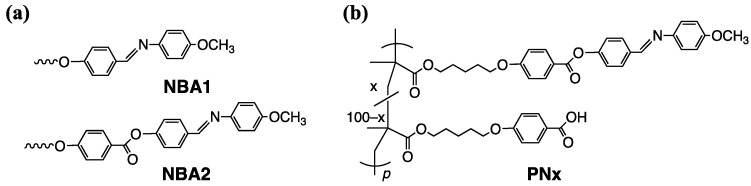
(**a**) Photoalignable mesogenic side groups. (**b**) Chemical structure of LC copolymers **PNx**.

**Figure 2 polymers-15-01408-f002:**
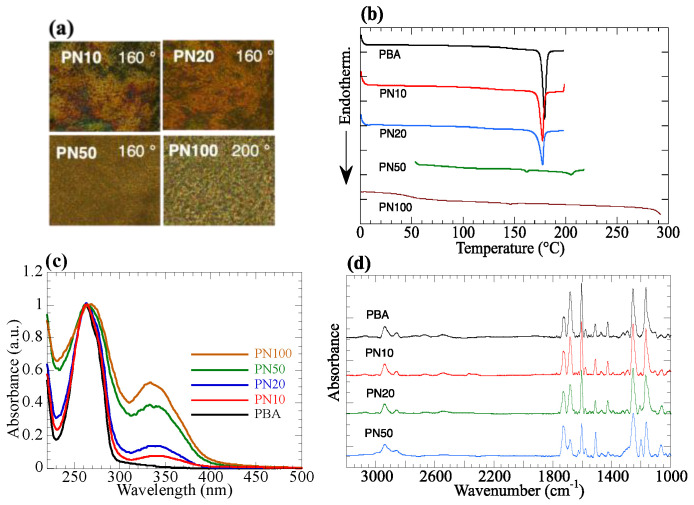
(**a**) POM photographs of LC copolymers. (**b**) DSC second heating curves of the copolymers. (**c**) UV absorption spectra of the copolymer films on quartz substrates. (**d**) FT-IR spectra of the copolymer films on CaF_2_.

**Figure 3 polymers-15-01408-f003:**
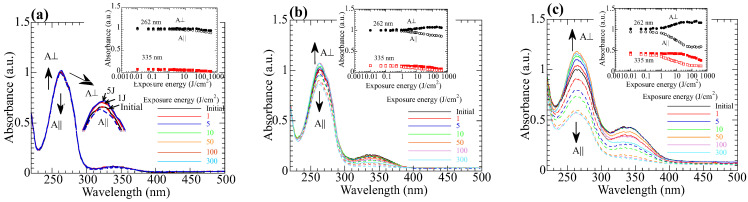
Changes in the polarized absorption spectra of the **PNx** films upon exposure to LP 365-nm light. Insets plot the absorbances at 262 and 335 nm. (**a**) **PN10**, (**b**) **PN20**, and (**c**) **PN50**.

**Figure 4 polymers-15-01408-f004:**
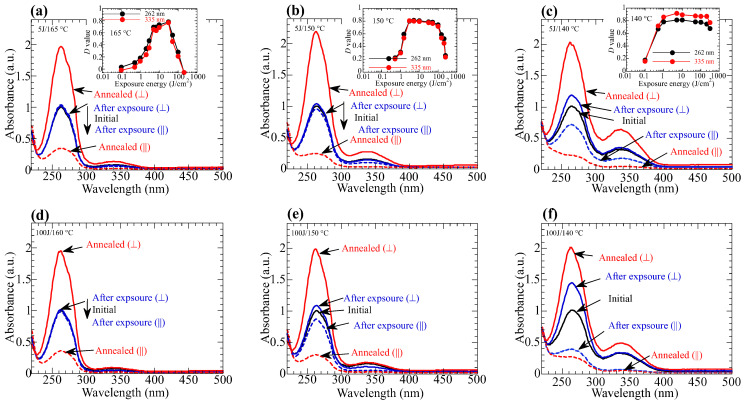
Changes in the polarized absorption spectra of the copolymer films before (black line) and after exposure (blue lines) to LP 365-nm light for (**a**–**c**) 5 J/cm^2^ and (**d**–**f**) 100 J/cm^2^, and subsequent annealing (red lines) at (**a**) 165 °C (**PN10**), (**d**) 160 °C (**PN10**), (**b**,**e**) 150 °C (**PN20**), (**c**,**f**) 140 °C (**PN50**). Inset plot changes in the *D* values as functions of exposure energy.

**Figure 5 polymers-15-01408-f005:**
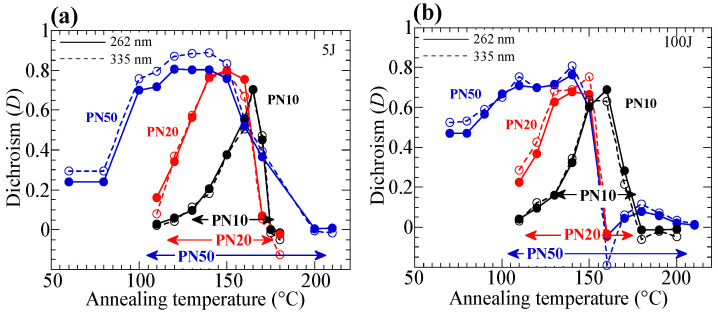
Changes in the thermally stimulated *D* values of the exposed **PNx** films as a function of the annealing temperature. Exposure doses are (**a**) 5 J/cm^2^, and (**b**) 100 J/cm^2^. Arrows indicate the LC temeperature range of **PNx**.

**Figure 6 polymers-15-01408-f006:**
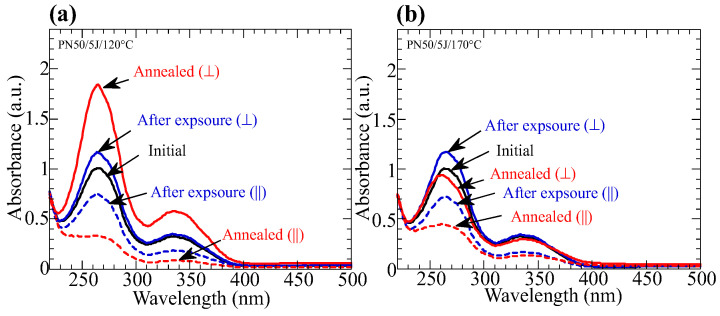
Changes in the polarized absorption spectra of the **PN50** films before (black line) and after exposure (blue lines) to LP 365-nm light for 5 J/cm^2^ and subsequent annealing (red lines) at (**a**) 120 °C, and (**b**) 170 °C for 10 min.

**Figure 7 polymers-15-01408-f007:**
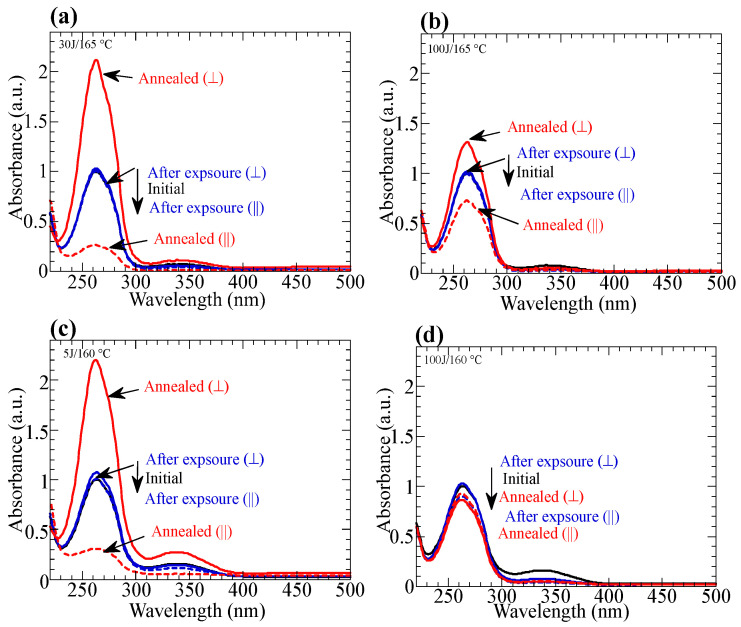
(**a**,**b**) Changes in the polarized absorption spectra of the **PN10** films before (black line) and after exposure (blue lines) to LP 365-nm light for (**a**) 30 J/cm^2^ and (**b**) 100 J/cm^2^ and subsequent annealing (red lines) at 165 °C for 10 min. (**c**,**d**) Changes in the polarized absorption spectra of the **PN20** films before (black line) and after exposure (blue lines) to LP 365-nm light for (**c**) 5 J/cm^2^ and (**d**) 100 J/cm^2^ and subsequent annealing (red lines) at 160 °C for 10 min.

**Figure 8 polymers-15-01408-f008:**
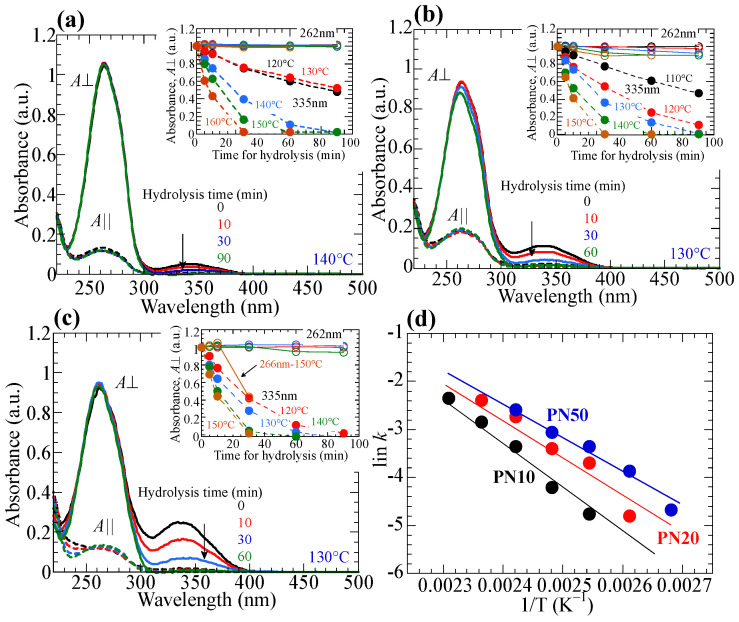
Changes in the polarized absorption spectra when oriented (**a**) **PN10**, (**b**) **PN20**, and (**c**) **PN50** films are annealed at 140 °C (**PN10**) or 130 °C (**PN20**, **PN50**) under RH = 100% atmosphere. Insets plot the changes in the normalized *A*_⏊_ values at 262 and 335 nm annealed at various temperatures under RH = 100% atmosphere as functions of the hydrolysis time. (**d**) Arrhenius plots of the hydrolysis of **PNx** films.

**Figure 9 polymers-15-01408-f009:**
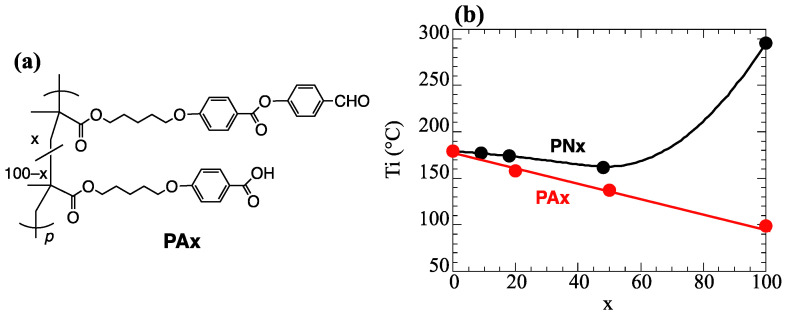
(**a**) Chemical structure of LC copolymers **PAx**. (**b**) T_i_ of copolymers as a function of copolymer composition.

**Figure 10 polymers-15-01408-f010:**
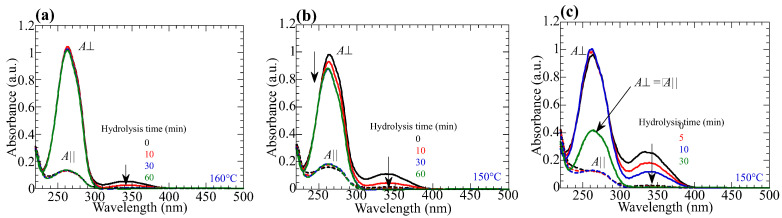
Changes in the polarized absorption spectra when oriented (**a**) **PN10**, (**b**) **PN20**, and (**c**) **PN50** films are annealed at 160 °C (**PN10**) or 150 °C (**PN20**, **PN50**) under RH = 100% atmosphere.

**Figure 11 polymers-15-01408-f011:**
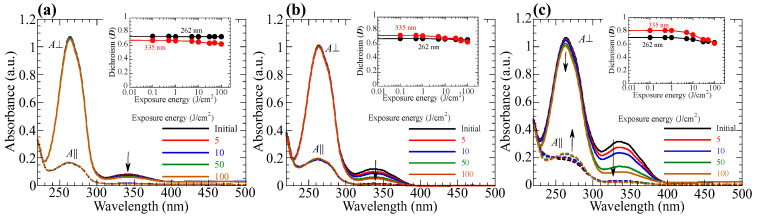
Changes in the polarized absorption spectra when oriented (**a**) **PN10**, (**b**) **PN20**, and (**c**) **PN50** films are exposed to LP 365-nm light with **E** parallel to the orientation direction. Insets plot the *D* values as a function of exposure energy.

**Figure 12 polymers-15-01408-f012:**
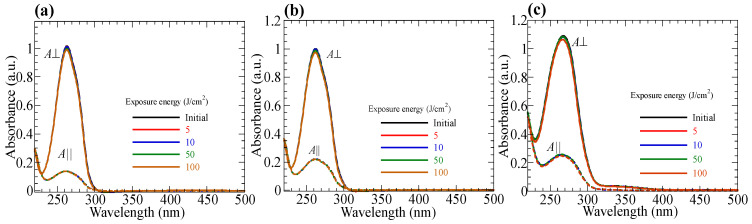
Changes in the polarized absorption spectra when hyrolyazed (**a**) **PN10**, (**b**) **PN20**, and (**c**) **PN50** films are exposed to LP 365-nm light with **E** parallel to the orientation direction.

**Table 1 polymers-15-01408-t001:** Composition, molecular weight, and thermal property of (co)polymers.

(Co)Polymers	X ^(a)^	Molecular Weight ^(b)^ Mn (Mw/Mn)	Thermal Property (°C) ^(c)^ (Transition Enthalpy, J/g)
**PBA**	0	22,000 (1.5)	G 137 N 179 (29.3) I
**PN10**	9	37,000 (2.6)	G 121 N 177 (24.6) I
**PN20**	18	28,000 (2.5)	G 112 N 174 (19.0) I
**PN50**	48	29,000 (2.1)	G 105 N_1_ 163 (0.63) N_2_ 205 (2.3) I
**PN100**	100	49,000 (1.9)	G 75 N 295 I

^(a)^ Determined by ^1^H-NMR. ^(b)^ Measured by GPC, ^(c)^ Determined by DSC and POM observations, G; glass, N; nematic, I; isotropic.

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
