# Peer review of "Photo-Durable Molecularly Oriented Liquid Crystalline Copolymer Film based on Photoalignment of N-benzylideneaniline"

_polymers, 2023, doi:10.3390/polym15061408_

Round 1

Reviewer 1 Report

The authors present a series of photo-aligned liquid crystalline copolymers. The UV irradiation influence on the alignment was investigated by several methods: IR, UV-vis spectroscopy. The possible application of such copolymers in befrigerant devices is presented. The overall presentation of the results is of high quality and I recommend to accept this manuscript after minor revisions:

The text in lines 217 - 221 have a different separation step than the others, it should be unified with the whole text.

The UV spectra in Fig. 2 should be described more thoroughly, how exactly this copolymer composition influences the UV?  What does this increasing band at 350 nm correspond to?

Author Response

Response to Reviewer 1

  • The text in lines 217 - 221 have a different separation step than the others, it should be unified with the whole text.

Answer:

We have divided the section to add Section 8, and revised the text to be in detail.

“3.7. Photo-durability of reoriented films”

“3.8. Photo-durability of hydrolyzed-oriented films

Hydrolyzed oriented films exhibit a higher photo-durability. Figures 12a–c show the changes in the polarized absorption spectra of a hydrolyzed oriented PN10, PN20, and PN50 films (DH > 95 %) upon the exposure to LP 365-nm light with E parallel to the orientation direction, respectively. The spectra of the oriented films do not change even after the exposure dose of 100 J/cm2 regardless of the composition. This is due to the lack of photo-sensitive moieties in the oriented films. These films are applicable to birefringent film for display application because there is no absorption in the visible light region.”

  • The UV spectra in Fig. 2 should be described more thoroughly, how exactly this copolymer composition influences the UV?  What does this increasing band at 350 nm correspond to?

Answer:

We revised the corresponding part as follows:

“ UV-vis spectra of the copolymer depend on the copolymer composition. They show the absorption band of BA at 262 nm and that of NBA2 at 335 nm (Fig. 2b). Because the absorption band at 262 nm overlaps with the NBA2 absorption, absorption maxima slightly shifts to the longer wavelength region as increase in the NBA2 composition, while the absorption maxima at 335 nm rarely changes due to no absorption of BA at 335 nm. FT-IR spectra show the vibration intensities of C=N (1624 cm–1), COOH (1683 cm–1), and H-bonded dimer (2681 and 2530 cm–1) also depend on the copolymer composition (Fig. 2c). These results indicate the random distribution of mesogenic side groups in the copolymers.”

Reviewer 2 Report

The authors developed photo-durable photoalignable liquid crystalline polymers and investigated its thermal and spectral properties, absorption, hydrolysis, and the photo-durability. The developed LC copolymer can maintain its oriented structure under UV light re-exposure. Such a non-rewritable photoalignable LC copolymer would benefit many applications. The manuscript is recommended to be published after addressing the following questions:

11.  Annealing at high temperature could promote an amplified anisotropy. Annealing under high humidity will induce hydrolysis of the oriented film, thus leading to a lower birefringence. Generally, an elevated temperature environment will have a lower relative humidity. Then why did the authors conduct the annealing process under a high humidity? The authors should clarify on it.  

22. The full names of several equipment used in the experiments are necessary, including GPC, DSC, and POM.

33.  In Sec. 3.2, the authors stated that the absorbance perpendicular to the polarization of the exposure light increases for all copolymers. As seen from the inset of Fig.3a, the absorbance of PN10 copolymer doesn't go up as the exposure energy increases.

44.  It is suggested to describe how to measure the absorbance.

55.  With regards to Fig. 6, the absolute value of absorbance at 262 nm and 335 nm is way different. The insets (Fig. 6) plot that both absorbance at 262 nm and 335 nm start to drop from 1. If the authors did the normalization, it should be mentioned in the revised manuscript for clarity.

Author Response

Response to Reviewer 2

  1. Annealing at high temperature could promote an amplified anisotropy. Annealing under high humidity will induce hydrolysis of the oriented film, thus leading to a lower birefringence. Generally, an elevated temperature environment will have a lower relative humidity. Then why did the authors conduct the annealing process under a high humidity? The authors should clarify on it.

Answer:

We added a detailed experimental procedure in the experimental section and text:

“Annealing at elevated temperatures under high humidity (RH>70 %) induced in situ hydrolysis of the oriented film. To attain high humidity at high environmental temperature, an oriented film was placed on a temperature-controlled stage placed in a box with high-temperature water. Adjusting the annealing time and its temperature controlled degree of the hydrolysis (DH), and DH was evaluated by the absorption band of NBA2 at 335 nm.”

Because the oriented films relatively stable when placed in the room, hydrolysis requires water and acid at high temperature.

  1. The full names of several equipment used in the experiments are necessary, including GPC, DSC, and POM.

Answer:

We added the full name of these.

  1. In Sec. 3.2, the authors stated that the absorbance perpendicular to the polarization of the exposure light increases for all copolymers. As seen from the inset of Fig.3a, the absorbance of PN10 copolymer doesn't go up as the exposure energy increases.

Answer:

It shows slight increase in perpendicular direction in PN10. Slight increase is due to a small amount of NBA2. We added an enlarged inset in Fig. 3a, and added the explanation in the text:

“The photoinduced optical anisotropy depends on the composition of NBA2 in the copolymer. Negative optical anisotropy is clearly seen in PN20 and PN50 due to large content of NBA2, while it is very small for PN10. The photoinduced reorientation of the mesogenic side groups is caused by the photoisomerization of NBA2 while it depends on the NBA2 composition.”

  1. It is suggested to describe how to measure the absorbance.

Answer:

Absorption spectra were measured by the UV-vis spectrometer as described in the experimental section. We added the following sentence in the experimental section.

“As a measure of the photoinduced optical anisotropy, polarization absorption spectra were measure using a Hitachi U-3010 spectrometer equipped with Glan-Taylor polarization prisms. The absorption spectrum in both parallel and perpendicular direction to the polarization (E) of the LP 365-nm was measured. The photoinduced in-plane dichroism (D) is estimated as “

  1. With regards to Fig. 6, the absolute value of absorbance at 262 nm and 335 nm is way different. The insets (Fig. 6) plot that both absorbance at 262 nm and 335 nm start to drop from 1. If the authors did the normalization, it should be mentioned in the revised manuscript for clarity.

Answer:

We added an explanation about this normalization in the Figure caption.

“Insets plot the changes in the normalized A^ values at 262 and 335 nm“

Reviewer 3 Report

Summary:  Watsuki and coworkers reported the photo-durability and reorientation of side-chain copolymers containing N-benzylideneaniline as a chromophore. The optical properties of photo-alignable copolymers were characterized by dichroism and birefringence measurements, and decomposition of the photosensitive moieties in the reoriented copolymer film was studied as photo-durability.

The authors reported the dichroism and birefringence values ​​of the copolymers. However, studies by other researchers for comparison were not included. Previous studies should be included in the Introduction section or text, especially dichroism and birefringence values.  

 Comments and questions:

1. In-plane dichroism (D) and birefringence are dependent on the sample thickness. This information should be included in the Experimental section.

2. To confirm the nematic phase of the samples, POM or X-ray data should be included.

3. Line 107-108: The authors mentioned a small transition enthalpy and high Tni of PN100. However, no nematic to isotropic (Tni) transition is observed in the DSC spectra of PN100 (Figure 2a). How were the authors able to determine the transition enthalpy and Tni value? Could PN100 start to decompose at ~300C? TGA measurement is needed for this sample.

4. The dichroism of 0.7 and birefringence of 0.113-0.118 of the copolymers were reported. These numbers do not seem very high. These values should be compared with other previous reports.

5. Figure 7 should be properly positioned on Page 8

Author Response

Response to Reviewer 3

The authors reported the dichroism and birefringence values ​​of the copolymers. However, studies by other researchers for comparison were not included. Previous studies should be included in the Introduction section or text, especially dichroism and birefringence values.  

Answer:

We revised introduction more detail.

Comments and questions:

  1. In-plane dichroism (D) and birefringence are dependent on the sample thickness. This information should be included in the Experimental section.

Answer:

We added the thickness of the sample for birefringence measurement.

“Birefringence of the 120–150 nm- thick oriented films was measured by a multi-channel polarization analyzer using Optipro-compact11MQ (Shintech Co. Ltd) at 607 nm”

  1. To confirm the nematic phase of the samples, POM or X-ray data should be included.

Answer:

We added the POM photos in the text.

  1. Line 107-108: The authors mentioned a small transition enthalpy and high Tni of PN100. However, no nematic to isotropic (Tni) transition is observed in the DSC spectra of PN100 (Figure 2a). How were the authors able to determine the transition enthalpy and Tni value? Could PN100 start to decompose at ~300C? TGA measurement is needed for this sample.

Answer:

Because the LC-isotropic transition of PN100 is partly accompanied by the degradation of the material when observing the POM. We added a sentence in the text.

“By contrast, the homopolymer with NBA2 side groups (PN100) displays a small transition enthalpy and high Ti, at which simultaneous decomposition of the material is seen in POM observation (Fig. 2a, Table 1)”

  1. The dichroism of 0.7 and birefringence of 0.113-0.118 of the copolymers were reported. These numbers do not seem very high. These values should be compared with other previous reports.

Answer:

We eliminated the word “very high”.

  1. Figure 7 should be properly positioned on Page 8

Answer:

We placed them properly.